# Assessing community-level impacts of and responses to stay at home orders: The King County COVID-19 community study

**Kathleen Moloney**[1]*, **Julio A. Lamprea Montealegre**[2], **Tania M. Busch Isaksen**[1], **Mallory Kennedy**[1], **Megan Archer**[1,3]☉, **Carlos Contreras**[1]☉, **Daaniya Iyaz**[1]☉, **Juliette Randazza**[1,4]☉, **Javier Silva**[1]☉, **Nicole A. Errett**[1,5]*

1 Department of Environmental and Occupational Health Sciences, University of Washington, Seattle, Washington, United States of America, 2 Division of Cardiology, University of California, San Francisco, San Francisco, California, United States of America, 3 College of Built Environments, University of Washington, Seattle, Washington, United States of America, 4 Evans School of Public Policy and Governance, University of Washington, Seattle, Washington, United States of America, 5 Department of Health Systems and Population Health, University of Washington, Seattle, Washington, United States of America

☉ These authors contributed equally to this work.
* kmoloney@uw.edu (KM); nerrett@uw.edu (NAE)

**Data Availability Statement:** We have created and published a project for this study on the National Science Foundation-funded DesignSafe-CI's Data

## Abstract

### Background

At the beginning of the COVID-19 pandemic, non-pharmaceutical interventions (NPIs) of unprecedented scope and duration were implemented to limit community spread of COVID-19. There remains limited evidence about how these measures impacted the lived experience of affected communities. This study captured the early impacts and coping strategies implemented in King County, Washington, one of the first U.S. communities impacted by COVID-19.

### Methods

We conducted a cross-sectional web-based survey of 793 English- and Spanish-speaking adult King County residents from March 18, 2020 –May 30, 2020, using voluntary response sampling. The survey included close- and open-ended questions on participant demographics, wellbeing, protective actions, and COVID-19-related concerns, including a freeform narrative response to describe the pandemic's individual-, family- and community-level impacts and associated coping strategies. Descriptive statistics were used to analyze close-ended questions, and qualitative content analysis methods were used to analyze free-form narrative responses.

### Results

The median age of participants was 45 years old, and 74% were female, 82% were White, and 6% were Hispanic/Latinx; 474 (60%) provided a qualitative narrative. Quantitative findings demonstrated that higher percentages of participants engaged in most types of COVID-19 protective behaviors after the stay-at-home order was implemented and schools

Depot Repository, and have uploaded the de-identified study dataset to the repository. In alignment with the repository's Protected Data Best Practices, this published dataset includes participants' responses to quantitative survey questions and the results of the qualitative data analysis of participants' written narratives (e.g., the binary '1' or '0' variable for each code to indicate if it was present in a participant's narrative). Participants' written narratives are not included in this publicly available dataset, as many contain sensitive and/or potentially identifiable information. As survey responses were collected from a small geographical area, sampling heavily from the University of Washington community, complete de-identification of participants' narratives is not possible. The de-identified data set is available for public access via DesignSafe project PRJ-2997 (DOI: https://doi.org/10.17603/ds2-atw6-7z47).

**Funding:** NAE was supported by the University of Washington Interdisciplinary Center for Exposures, Diseases, Genomics and Environment (National Institute of Environmental Health Sciences, https://www.niehs.nih.gov/; Grant #: P30ES007033). REDCap at ITHS is supported by the National Center For Advancing Translational Sciences (https://ncats.nih.gov/) of the National Institutes of Health under Award Number UL1 TR002319. JLM is supported by National Heart, Lung, and Blood Institute (https://www.nhlbi.nih.gov/; Grant #: 1K99HL157721-01A1). The content is solely the responsibility of the authors and does not necessarily represent the official views of the National Institutes of Health. The funders had no role in study design, data collection and analysis, decision to publish, or preparation of the manuscript.

**Competing interests:** The authors have declared that no competing interests exist.

and community spaces were closed, relative to before, and that participants tended to report greater concern about the pandemic's physical health or healthcare access impacts than the financial or social impacts. Qualitative data analysis described employment or financial impacts (56%) and vitality coping strategies (65%), intended to support health or positive functioning.

## Conclusions

This study documented early impacts of the COVID-19 pandemic and the NPIs implemented in response, as well as strategies employed to cope with those impacts, which can inform early-stage policy formation and intervention strategies to mitigate the negative impacts. Future research should explore the endurance and evolution of the early impacts and coping strategies throughout the multiyear pandemic.

## Introduction

The impacts of COVID-19 and associated societal responses are multidimensional and an area of current research. Moreover, these impacts have been shown to be dynamic and community specific [1, 2]. Non-pharmaceutical interventions (NPIs), including masking, social distancing, and community-wide shutdowns, were one form of societal response that was widely implemented throughout the pandemic. Even following development and widespread availability of effective medical countermeasures, NPIs remain an important contributor to controlling community transmission, particularly in the face of variants of concern [3]. However, the impacts of NPIs on mental health, access to health care, financial stability, and other important aspects of wellbeing require further study for a more holistic understanding of their consequences.

One type of NPI, stay-at-home orders, shutdowns, or lockdowns, have been implemented by governments across the world to curb the spread of SARS-CoV-2 and associated COVID-19 [4, 5]. Though stay-at-home orders have demonstrated effectiveness in lowering rates of SARS-CoV-2 transmission and preventing the excess morbidity and mortality associated with COVID-19, this type of NPI also has unintended negative impacts [6]. For instance, these orders, which forced closure of "non-essential" businesses, had various impacts on work. It is estimated that approximately 9.6 million U.S. workers lost their employment during 2020 as a result of COVID-19-related employment disruptions, a disproportionate share of which were low-wage workers [7]. In low- and middle-income countries, economic recessions, such as the initial recession caused by the COVID-19 pandemic, are associated with excess mortality. For example, a survey of over 30,000 participants from Burkina Faso, Ghana, Kenya, Rwanda, Sierra Leone, Bangladesh, Nepal, Philippines, and Colombia found massive increases in food insecurity as a result of the economic insecurity created by the pandemic and associated NPIs [8].

Many schools were also closed for months as part of the stay-at-home orders, impacting the academic outcomes of students worldwide. A systematic review of the impacts of COVID-19-related school closures on student achievement estimated that primary school children's reading and mathematics test scores decreased by a median of 0.37 standard deviations during remote learning, and that the academic achievement gap between students of high and low socioeconomic status would increase by nearly 30% [9]. Many students unenrolled as a result of the school closures, with potentially disproportionate impacts in the global South; a study

conducted in Ethiopia, Malawi, Nigeria, and Uganda found that enrollment of children in primary school dropped from 96% to around 46% during the pandemic [10].

The COVID-19 pandemic and responsive stay-at-home orders are also associated with negative physical health impacts beyond the health consequences of infection with SARS-CoV-2. For example, reduced physical activity has been shown to be aligned with implementation of stay-at-home orders [11, 12]. Additionally, access to healthcare was limited by the burden the pandemic placed on many healthcare systems, leading to delays in care for patients with chronic and emergent health conditions and a temporary suspension of preventive healthcare, such as cancer screenings. Research continues to emerge about the excess morbidity and mortality likely attributable to these delays in care. A study in the United Kingdom, for example, estimated between a 4.8 to 16.6 increase in deaths due to various types of cancers as a result of delayed screenings during the pandemic [13].

The mental health impacts of the pandemic are equally alarming. COVID-19 stay-at-home orders have been associated with greater health anxiety, financial worry, and loneliness [14]. For instance, health-related quality of life has declined from pre-pandemic levels, particularly among young adults [15]. Healthcare workers in particular have suffered many adverse mental health impacts, with 41% of healthcare workers enrolled in the HERO study from April to July 2020 expressing feelings of burnout [16]. Racial and ethnic minorities, individuals with disabilities, women, and those who are precariously housed or employed have borne a disproportionate burden across all categories of the pandemic's impacts, including risk of COVID-19 exposure, additional health-related consequences, and disruptions to the social and economic underpinnings of health and wellbeing [17, 18].

While national and international surveys have attempted to understand the immediate impacts of NPIs on individuals and households [19–21], less research has explored such impacts in a geographically bounded community. Given the variability of NPIs implemented in response to COVID-19 between various U.S. states and counties, research examining the impacts on a smaller geographical area where all community members experienced the same COVID-19 preventive measures is warranted. Moreover, research has shown that implementation of the same NPIs have resulted in different impacts in different contexts. For instance, our own research on bike and pedestrian trail use changes following COVID-19 stay-at-home orders in Seattle, Houston, and New York found that while there were significant impacts in all cities, the direction of change varied by location [12]. Accordingly, the King County COVID-19 Community Study (KC3S) aims to learn from the experiences of the residents of a single county, King County, WA.

Washington was the first U.S. state known to be affected by COVID-19. In response, the state and King County, home to the city of Seattle, were among the first to implement community-level NPIs to stem SARS-CoV-2 transmission. At various stages of the pandemic, these strategies consisted of NPIs such as closures for certain businesses; a statewide Stay Home, Stay Healthy order; and mask mandates [22]. Additionally, public schools, colleges, and universities in King County largely offered online-only instruction, and many local health care providers suspended elective procedures and discouraged patients from seeking in-person, non-emergency care during the first year of the pandemic.

The King County COVID-19 Community Study compliments the several national and international surveys that have been distributed in response to COVID-19 by providing accounts of impacts and resilience in the context of community-specific COVID-19 impacts and implementation of NPIs. This exploratory, descriptive study is guided by two central research questions: 1) How are individuals, families, and communities impacted by COVID-19 and responsive NPIs implemented in their community?, and 2) How are individuals, families, and communities adapting to and coping with the impacts of COVID-19 and responsive

NPIs implemented in their community [23]? The King County COVID-19 Community Study aimed to characterize, in real time, how the NPIs implemented in King County impacted the lives and wellbeing of individuals, families, and communities, as well as the ways in which individuals, families, and communities have adapted and coped in the initial phases of the pandemic.

## Methods

We conducted an online survey of King County community members to understand the impacts of COVID-19 and responsive NPIs on individuals, their families, and broader community, as well as associated coping mechanisms, in the initial phase of the COVID-19 pandemic response using a voluntary response sample approach [23]. Study data were collected and managed using REDCap (Research Electronic Data Capture) tools [24] hosted at the Institute of Translational Health Sciences. Research Electronic Data Capture is a secure, web-based application designed to support data capture for research studies, providing: 1) an intuitive interface for validated data entry; 2) audit trails for tracking data manipulation and export procedures; 3) automated export procedures for seamless data downloads to common statistical packages; and 4) procedures for importing data from external sources.

The survey included a mix of multiple choice and open-ended questions about experiences with COVID-19, engagement in behaviors to minimize COVID-19 risk, level of concern with potential impacts of COVID-19, emotional state, presence of COVID-19 symptoms, impacts of COVID-19 on work, and use of tools to cope with COVID-19-related stress and other impacts [23]. The World Health Organization-Five Well-Being Index (WHO-5), a brief measure with 5 items rated on a 0 to 5-point Likert scale, was included in the survey as a validated measure of current mental wellbeing [25]. The total sum of all 5 items is converted into a percentage score, with 0 representing the worst possible wellbeing and 100 representing the best possible wellbeing; a score of 50 or less is generally used as a cutoff when screening for clinical depression [25]. English and Spanish versions of the WHO-5 questionnaire are publicly available at https://www.psykiatri-regionh.dk/who-5/who-5-questionnaires/Pages/default.aspx [26]. The survey also collected information about respondents' demographic characteristics, including age, gender identity, education, income, race/ethnicity, health insurance status, and area residence (e.g., ZIP code, county, neighborhood), and households (e.g., number of people overall, under 18, and high-risk for COVID-19 in the respondent's household). Survey items without a pre-existing validated Spanish translation were translated by a study team member who was a native Spanish-speaker, also fluent in English. Prior to the release of the survey, multiple members of the research team with a variety of education and experience levels reviewed the survey and piloted taking the survey via the REDCap form to ensure accessibility and readability.

The University of Washington Human Subjects Division (HSD) reviewed the study protocol and determined the study was human subjects research that qualified for exempt status (STUDY00009765). Participants were provided an overview of the potential risks and benefits of study participation on the study webpage, and informed that accessing the survey via the provided link and answering survey questions indicated consent to participate in the research study.

### Study setting

King County, which includes the city of Seattle, has a total population of just under 2.3 million people [27]. Approximately 64% of the population identifies as White, 7% as Black or African American, 1% as American Indian or Alaska Native, 22% as Asian, 1% as Native Hawaiian or

Pacific Islander, and 6% as mixed race [27]. Eleven percent of the population identifies as Hispanic or Latinx, and approximately 7% of the population speaks Spanish [28]. The median household income is $106,326 [27]. Of those aged 25 or older, approximately 94% have at least a high school degree and 54% have at least a Bachelor's degree [27].

## Sample and data collection

To participate in the survey, respondents were required to indicate in the survey that they were at least 18 years old, living in King County, and able to respond to the survey in the languages offered (Spanish and English). Only those who indicated an age less than 18 years or that they did not reside in King County were excluded from study participation. The research team employed a multipronged approach to recruit community members to complete the survey. We posted information about the survey through community-based organizations, University of Washington websites, social media, online forums (e.g., Next-door), flyers placed in communities and businesses, local media organizations, and word of mouth. Recruitment materials described the study and directed interested participants to a website to learn more about the study and participate. Respondents completed the survey between March 18, 2020, and May 30, 2020 [23].

## Participant characteristics

In total, 793 participants completed or partially completed the online survey, with 787 (99%) participants completing the survey in English and 6 (1%) participants completing the survey in Spanish. The participants ranged in age from 18 to 87 years old, with a median age of 45 (IQR 33–56, Table 1). Seventy-four percent of the sample identified as female, 20% as male, and 3% as non-binary. Most participants had attended at least some college, with 39% of participants having attained a bachelor's degree and 43% of participants having attained a graduate degree. Over two-thirds of participants were currently employed at the time of survey completion (55% full-time and 14% part-time); 12% of participants were retired and 3% were currently unemployed.

## Data analysis

The survey data were exported to Microsoft Excel for review and cleaning. Once the data were cleaned, the research team calculated descriptive statistics for multiple choice questions. Differences in participants' reported frequencies of engaging in COVID-19 protective behaviors, such as increased handwashing, decreased use of public transport, or avoidance of gatherings, before and after March 15, 2020, the date when K-12 schools as well as bars, restaurants and places of recreation were ordered to close in King County, were analyzed using McNemar's test for equality of paired proportions. A WHO-5 Well-being Index percentage score was calculated for the subset of participants who wrote a narrative description of their experiences during the COVID-19 pandemic, to allow for comparison of these scores to participants' open-ended descriptions of impacts and coping strategies. Differences in WHO-5 Well-being Index percentage scores across gender and age groups of participants were analyzed using the Kruskal-Wallis rank sum test.

To analyze the open-ended data detailing respondents' stories about COVID-19, we used a directed content analysis approach [29]. A codebook was developed based on the study goals, and the framework for describing disaster losses and coping strategies presented in a recently published disaster research article by Peek et al [30]. One member of the study team applied the codebook to a sample of open-ended responses to assess the robustness of the codebook relative to the data prior to broader application by the study team. The codebook included

**Table 1. General characteristics of participants (N = 793).**

| Characteristic | |
|---|---|
| **Age, years**; median (IQR)[a] | 45 (33–56) |
| **Gender**; n (%) | |
| Female | 583 (74) |
| Male | 156 (20) |
| Non-binary | 22 (3) |
| Other | 3 (>1) |
| Missing | 29 (4) |
| **Race**; n (%) | |
| White | 650 (82) |
| Black | 13 (2) |
| Asian | 63 (8) |
| American Indian or Alaska Native | 11 (1) |
| Native Hawaiian or Pacific Islander | 4 (1) |
| Other | 42 (5) |
| Missing | 10 (1) |
| **Ethnicity**; n (%) | |
| Non-Hispanic/Non-Latinx | 737 (93) |
| Hispanic/Latinx | 45 (6) |
| Missing | 11 (1) |
| **Employment**; n (%) | |
| Full time | 434 (55) |
| Part time | 109 (14) |
| Retired | 99 (12) |
| Unemployed | 20 (3) |
| Other | 126 (16) |
| Missing | 5 (1) |
| **Education**; n (%) | |
| High school | 16 (2) |
| Some college | 80 (10) |
| Associates degree | 43 (5) |
| Bachelor's degree | 306 (39) |
| Graduate degree | 342 (43) |
| Missing | 6 (1) |
| **Gross annual household income**, USD; median (IQR) | 110,000 (68,000–180,000) |
| **Number of household members**; median (IQR) | 2 (2–3) |
| **Number of household members <18 years old**; median (IQR) | 0 (0–1) |
| **Health insurance**; n (%) | 770 (97) |
| **Spanish survey**; n (%) | 6 (1) |
| **COVID-19 symptoms & testing**; n (%) | |
| Experienced flu-like symptoms | 225 (28) |
| Tested for COVID-19 | 15 (2) |
| Positive COVID-19 test | 11 (1) |

[a] IQR: interquartile range

code names, definitions, examples of correct code application from the survey responses, and notes about inclusion and exclusion criteria. One member of the research team trained the other coders on the codebook. The group engaged in a consensus-building activity that

consisted of applying the draft codebook to 20 stories independently, meeting to discuss and adjudicate differences in applied codes, and revising the codebook in accordance with group discussion. Codes and definitions from the final codebook are provided in Table 2 below.

To reduce bias, after the initial round of coding, three teams of two coders each worked together to co-code the remainder of the open-ended responses using NVivo qualitative analysis software. The coders worked in groups of two to discuss and adjudicate differences in their coding. Coders posed questions about the coding to the larger group via email as needed.

Upon completing coding, the team merged the coding teams' NVivo files into a single file. The qualitative data was then merged with the quantitative data file, with each code included as a binary variable coded as a '1' if it was present in each participant's story, and '0' if it was not present. The percentage of participants' stories that included each code was calculated, and differences between gender and age groups were compared using Fisher's exact test. Participants who did not provide a response to the open-ended prompt were not included in this analysis. Associations between codes describing COVID-19-related impacts and codes describing coping strategies were also examined. All qualitative data analysis was completed in NVivo Version 12 and NVivo for Mac [31]; quantitative analyses were performed using RStudio Version 4.1.2 [32].

**Table 2. Qualitative codes and illustrative quotes from KC3S survey participants.**

| Code | Definition |
| --- | --- |
| **Impacts at the Individual, Family & Community Level** | |
| *Employment or Financial Impact* | Impact to employment or finances |
| *Covid-19 Physical Health Impact* | Covid-19-related symptoms or death |
| *Other Physical Health Impact* | Impact to physical health, including disruptions to health care |
| *Mental or Behavioral Health Impact* | Impact to mental health, including disruptions to mental health care |
| *Social Impact* | Impact to social life |
| *Education or Childcare Impact* | Impact to childcare or education (elementary school, high school, higher education, etc.) |
| *Inequitable Impact* | Impact that disproportionately affects a group or groups |
| *Displacement Impact* | Impact to location or housing |
| *Lasting Impact* | Impacts that respondent thinks or hopes will be lasting |
| *Other Impact* | Impacts that do not fall into one of the above categories |
| *No Impact* | Absence of impact |
| **Coping Strategies at the Individual, Family & Community Level** | |
| *Vitality Coping Strategy* | Strategy intended to support health or positive functioning |
| *Opportunity Strategy* | Strategy intended to support the achievement of life goals or financial stability |
| *Connectedness Strategy* | Strategy intended to maintain or strengthen social/community support and interdependence |
| *Contribution Strategy* | Strategy intended to support meaning, purpose, engagement, or belonging |
| *Inspiration Strategy* | Strategy intended to support motivation or hopefulness |
| *No Individual Coping Strategy Needed* | Lack of need for coping strategy |
| **Perspective on Community & Government Response** | |
| *Inadequate Community Response* | Failure of community to respond |
| *Inadequate Government Response* | Failure of systems; mistrust of federal or other government leaders or systems; questioning government decisions/strategies; lack of political will or ethical leadership; inadequate policy support |

To allow for accurate assessment of areas of convergence and divergence between the quantitative and qualitative data, we also conducted a sensitivity analysis of the quantitative survey data capturing participants' level of concern about potential impacts of the pandemic; only participants who responded to both the survey questions and provided an open-ended narrative were included in this analysis.

## Results

### Engagement in COVID-19 protective behaviors

Participants were asked to indicate if they had increased engagement in specific behaviors to minimize COVID-19 risk both prior to March 15th, 2020 and after March 15th, 2020. Across nearly every category of COVID-19 protective behavior, a statistically significantly higher percentage of participants ($p < 0.05$) reported engaging in the behavior after March 15th, 2020, relative to prior to that date (Table 3). The three exceptions were: "avoided large gatherings", specified as greater than 50 people (82% vs 80%, p = 0.2); "increased handwashing for at least 20 seconds with soap and water"(88% vs 81%, p<0.001); and "increased hand sanitizer use"(65% vs 58%, p<0.001), perhaps due to earlier public health recommendations regarding these protective actions [33].

### Degree of concern about potential COVID-19 impacts

Participants were also asked to rate the degree of their concern about various potential COVID-19 impacts, which included impacts to physical health, financial stability, and social isolation (Table 4). In general, participants reported a greater degree of concern over the potential physical health or healthcare access impacts of the pandemic than the potential financial or social impacts. The highest percentage of participants reported being "very concerned" about "an at-risk family member getting sick" (69%), followed by "not being able to receive healthcare" (39%). Around half of participants reported being "somewhat concerned" about

**Table 3. Reported frequencies of COVID-19 protective behaviors before and after March 15th, 2020[a].**

| | n (%) | | |
|---|---|---|---|
| **Behavior** | **Before 3/15/20** | **After 3/15/20** | **P-value [b]** |
| Increased handwashing for at least 20 seconds with soap and water | 700 (88) | 644 (81) | <0.001 |
| Increased hand sanitizer use | 512 (65) | 457 (58) | <0.001 |
| Stayed home more often | 564 (71) | 685(86) | <0.001 |
| Increased time working from home | 394 (50) | 504 (64) | <0.001 |
| Decreased or discontinue going to bars and restaurants | 497 (63) | 652 (82) | <0.001 |
| Decreased or discontinue going to gym | 387 (48) | 498 (63) | <0.001 |
| Increased outdoor physical activity | 281 (35) | 427 (54) | <0.001 |
| Decrease or discontinued use of public transportation | 402 (51) | 453 (57) | <0.001 |
| Avoided small gatherings (<10 people) | 343 (43) | 700 (88) | <0.001 |
| Avoided midsize gatherings (11–50 people) | 567 (72) | 675 (85) | <0.001 |
| Avoided large gatherings (>50 people) | 634 (80) | 653 (82) | 0.2 |
| Canceled or delayed routine healthcare appointment or elective procedure | 270 (34) | 436 (55) | <0.001 |
| Canceled or delayed routine dental appointment | 241 (30) | 393 (50) | <0.001 |
| Purchased extra food or commodities | 451 (57) | 510 (64) | <0.001 |

[a] The date when King County K-12 schools, bars, restaurants and places of recreation were ordered to close
[b] P-value based on McNemar's test for equality in paired proportions

**Table 4. Reported concern about potential COVID-19 impacts.**

| | n (%) | | |
|---|---|---|---|
| | **Not concerned** | **Somewhat concerned** | **Very concerned** |
| **Getting sick myself** (n = 789) | 113 (14) | 444 (56) | 232 (29) |
| **An at-risk family member getting sick** (n = 772) | 45 (6) | 198 (26) | 529 (69) |
| **Not being able to receive healthcare** (n = 781) | 151 (19) | 327 (42) | 303 (39) |
| **Not being able to access healthcare for conditions other than Covid-19** (n = 778) | 357 (46) | 262 (34) | 159 (20) |
| **Not being able to access healthcare for emergent conditions other than Covid-19** (n = 783) | 192 (25) | 358 (46) | 233 (30) |
| **Not being able to work** (n = 760) | 382 (50) | 200 (26) | 178 (23) |
| **Not being able to pay bills** (n = 770) | 409 (53) | 185 (24) | 176 (23) |
| **Inability to retire on time** (n = 742) | 495 (67) | 140 (19) | 107 (14) |
| **Being socially isolated myself** (n = 778) | 325 (42) | 312 (40) | 141 (18) |
| **My family members being socially isolated** (n = 776) | 172 (22) | 368 (47) | 236 (30) |

"getting sick myself" (56%), "my family members being socially isolated" (47%), and "not being able to access healthcare for emergent conditions other than COVID-19" (46%). A majority of respondents reported being "not concerned" about "inability to retire on time" (67%), "not being able to pay bills" (53%), and "not being able to work" (50%). The ranking of concerns remained consistent when only data from participants who provided a qualitative narrative was included, with the distribution of participants responding "not concerned", "somewhat concerned", or "very concerned" to each item differing by at most 1–3% from the overall sample.

## WHO-5 well-being index scores

We calculated WHO-5 Well-being Index percentage scores for the subset of participants who wrote a narrative description of their experiences during the COVID-19 pandemic (n = 474). The median WHO-5 Well-being Index percentage score amongst the overall sample was 72 (interquartile range (IQR) 56–88). There was no statistically significant difference in median WHO-5 Well-being Index percentage scores by gender. However, statistically significantly lower WHO-5 Well-being Index percentage scores were associated with higher age ($p<0.001$); those in the eldest age group (65 years and older) had a median score of 60 (IQR 48–80), versus a median score of 76 (IQR 64–88) in the youngest age group (18 to 34 years old).

## Open-Ended narratives about COVID-19 experiences

A total of 474 survey participants (60%) wrote a narrative description of their experiences during the COVID-19 pandemic, including impacts to themselves, their family, and/or their broader community and associated coping strategies. These 474 narratives were included in qualitative data analysis. Qualitative codes included in the final version of the codebook described three general categories of participants' experiences: impacts of the COVID-19 pandemic or the associated NPIs, coping strategies employed to mitigate the impacts, and perspectives about the community or government response to the pandemic (Table 2).

## Impacts at the individual, family, and community level

Among all participants who provided a written narrative, the most frequently reported impact of the pandemic was "Employment or Financial Impact," with 56% of participants' narratives describing this type of impact (Table 5). Examples of the employment or financial impacts described by participants included "being the sole source of income in my family due to being

**Table 5. Impacts of covid-19 lockdowns and coping strategies of qualitative study participants by gender and age group.**

| Variable | Overall | Gender [a] | | | | Age Group (years) [a] | | | | |
|---|---|---|---|---|---|---|---|---|---|---|
| | N = 474 | Female, n = 342 | Male, n = 97 | Non-binary, n = 17 | p-value [b] | 18–34, n = 124 | 35–54, n = 221 | 55–64, n = 77 | 65+, n = 52 | p-value [b] |
| **WHO-5 Well-being Index Percentage Score**; median (IQR) [c] | 72 (56–88) | 72 (56–88) | 68 (56–84) | 84 (60–100) | 0.5 | 76 (64–88) | 72 (60–88) | 66 (52–88) | 60 (48–80) | **<0.001** |
| **Impacts at the Individual, Family & Community Level;** n (%) | | | | | | | | | | |
| Employment or Financial Impact | 264 (56) | 194 (57) | 57 (59) | 8 (47) | 0.6 | 75 (60) | 130 (59) | 44 (57) | 15 (29) | **<0.001** |
| Social Impact | 183 (39) | 140 (41) | 33 (34) | 4 (24) | 0.2 | 52 (42) | 72 (33) | 33 (43) | 25 (48) | 0.10 |
| Other Physical Health Impact | 181 (38) | 142 (42) | 24 (25) | 9 (53) | **0.006** | 42 (34) | 79 (36) | 34 (44) | 26 (50) | 0.13 |
| Mental or Behavioral Impact | 167 (35) | 129 (38) | 23 (24) | 6 (35) | **0.012** | 45 (36) | 81 (37) | 22 (29) | 18 (35) | 0.6 |
| Education or Childcare Impact | 113 (24) | 89 (26) | 18 (19) | 2 (12) | 0.2 | 24 (19) | 71 (32) | 15 (19) | 3 (6) | **<0.001** |
| Other Impact | 89 (19) | 66 (19) | 18 (19) | 4 (24) | 0.6 | 20 (16) | 40 (18) | 16 (21) | 13 (25) | 0.5 |
| No Impact | 68 (14) | 46 (13) | 14 (14) | 2 (12) | >0.9 | 18 (15) | 40 (18) | 6 (8) | 4 (8) | 0.066 |
| Covid-19 Physical Health Impact | 27 (6) | 19 (6) | 7 (7) | 1 (6) | ** | 9 (7) | 13 (6) | 4 (5) | 1 (2) | ** |
| Inequitable Impact | 25 (5) | 21 (6) | 4 (4) | 0 (0) | ** | 3 (2) | 13 (6) | 4 (5) | 5 (10) | ** |
| Lasting Impact | 20 (4) | 16 (5) | 2 (2) | 1 (6) | ** | 5 (4) | 8 (4) | 2 (3) | 5 (10) | ** |
| Displacement Impact | 12 (3) | 11 (3) | 1 (1) | 0 (0) | ** | 6 (5) | 5 (2) | 1 (1) | 0 (0) | ** |
| **Coping Strategies at the Individual, Family & Community Level;** n (%) | | | | | | | | | | |
| Vitality Coping Strategy | 306 (65) | 228 (67) | 56 (58) | 13 (76) | 0.2 | 73 (59) | 134 (61) | 55 (71) | 43 (83) | **0.007** |
| Connectedness Strategy | 150 (32) | 121 (35) | 21 (22) | 4 (24) | **0.008** | 37 (30) | 59 (27) | 29 (38) | 24 (46) | **0.030** |
| Contribution Strategy | 76 (16) | 62 (18) | 11 (11) | 2 (12) | 0.4 | 15 (12) | 31 (14) | 18 (23) | 12 (23) | 0.071 |
| Inspiration Strategy | 64 (14) | 52 (15) | 9 (9) | 1 (6) | 0.13 | 14 (11) | 23 (10) | 15 (19) | 12 (23) | **0.034** |
| Opportunity Strategy | 26 (5) | 22 (6) | 3 (3) | 1 (6) | ** | 8 (6) | 11 (5) | 5 (6) | 2 (4) | ** |
| No Individual Coping Strategy Needed | 3 (1) | 1 (0) | 2 (2) | 0 (0) | ** | 1 (1) | 2 (1) | 0 (0) | 0 (0) | ** |
| **Perspective on Community & Government Response;** n (%) | | | | | | | | | | |
| Inadequate Government Response | 73 (15) | 55 (16) | 13 (13) | 1 (6) | 0.3 | 18 (15) | 30 (14) | 14 (18) | 11 (21) | 0.5 |
| Inadequate Community Response | 66 (14) | 52 (15) | 8 (8) | 3 (18) | **0.009** | 21 (17) | 34 (15) | 8 (10) | 3 (6) | 0.2 |

[a] 16 participants missing gender data and 1 participant missing age data were excluded

[b] Kruskal-Wallis rank sum test; Fisher's exact test; results significant at the p <0.05 level are bolded; p-values for impacts and coping strategies endorsed by less than 30 participants from the overall sample were removed

[c] IQR: interquartile range

the only one who can work remotely", "COVID-19 has almost halted my business completely", and "friends who were sent home without pay for at least two to three weeks." One participant described losing their job due to contracting what they believed was COVID-19, stating they had been "[laid] off because [they] didn't get better."

"Social Impact" (39% of participants) was the second most frequently reported impact. As one participant wrote, describing how their pre-existing mental health conditions made coping with the social isolation created by stay-at-home orders particularly difficult,

> *"My history of PTSD [post-traumatic stress disorder] and Depression make isolation especially hard…I feel terrified at the idea that no one can hold me. I live alone and have no physical touch."*

"Other Physical Health Impact" (38%), such as being unable to "receive [medical] treatment for the foreseeable future" for chronic health conditions, and "Mental or Behavioral Impact"

(35%), such as an inability to be "productive at all", were the next most frequently cited impacts. Around 14% of participants described the pandemic as currently having no impact on their lives (Table 5). One such participant, describing how their partner's ability to work remotely had insulated them from some of the pandemic's impacts, stated,

> "My husband's company was one of the first to send people home [on March 2nd]. His ability to earn a paycheck and have satisfying work and be safe during this time is an immense relief."

Comparing the prevalence of impacts by gender, statistically significant differences were found for two impact categories, "Other Physical Health Impact" (p = 0.006) and "Mental or Behavioral Impact" (p = 0.012). Those who identified as female or non-binary were more likely than those who identified as male to describe experiencing each of these types of impacts (Table 5). Comparing the prevalence of impacts by age group, those in the youngest age group (18 to 34 years old) were the most likely to express an employment or financial impact (60%), versus only 29% of those in the eldest age group, 65 years and older (p<0.001). Those in the 35 to 54 years old group were the most likely to express an education or childcare impact (32%), as compared with 6% of the 65 years and older group and 19% of both the 18 to 34 years old and 55 to 64 years old groups (p<0.001, Table 5).

## Coping strategies at the individual, family, and community level

Among all participants included in qualitative data analysis, the most frequently employed coping strategy in response to the pandemic's impacts was "Vitality Coping Strategy"; 65% of participants' narratives described using this type of coping strategy (Table 5). Examples of the vitality coping strategies described by participants include "trying to educate our children on self care", "cleaning, sanitizing, stopping visitors, stocking up on supplies", and "still going to trails for hike or bike if not too crowded." "Connectedness Strategy" (32% of participants) was the next most frequently applied coping strategy code. Examples of the connectedness coping strategies described by participants include "using Zoom with my friends to create social gatherings", "increased talking to friends and family via phone", and "taking photos and videos to post to a group for the neighborhood on Facebook." Only 1% of participants stated that they did not need any individual coping strategy. As one such participant stated, the stay-at-home order had not created a major shift in their usual, pre-pandemic routine:

> "Thankfully, I've been working from home for 11.5 years and am an introvert, so staying at home hasn't been a huge adjustment for me."

Comparing the prevalence of coping strategies by gender, statistically significant differences were found for only "Connectedness Strategy" (p = 0.008). Those who identified as female were more likely than those who identified as male or non-binary to describe employing this type of coping strategy (Table 5). Comparing the prevalence of coping strategies by age group, three types of coping strategies were found to be statistically significantly different between groups, "Connectedness Strategy" (p = 0.03), "Vitality Coping Strategy" (p = 0.007), and "Inspiration Strategy" (p = 0.034). For each of these coping strategies, being a member of an older age group was associated with increased prevalence of employing the coping strategy.

## Associations between impacts and coping strategies

Distributions of types of coping strategies associated with each category of impact are shown in Fig 1. Regardless of the type of impact a participant described, they were most likely to

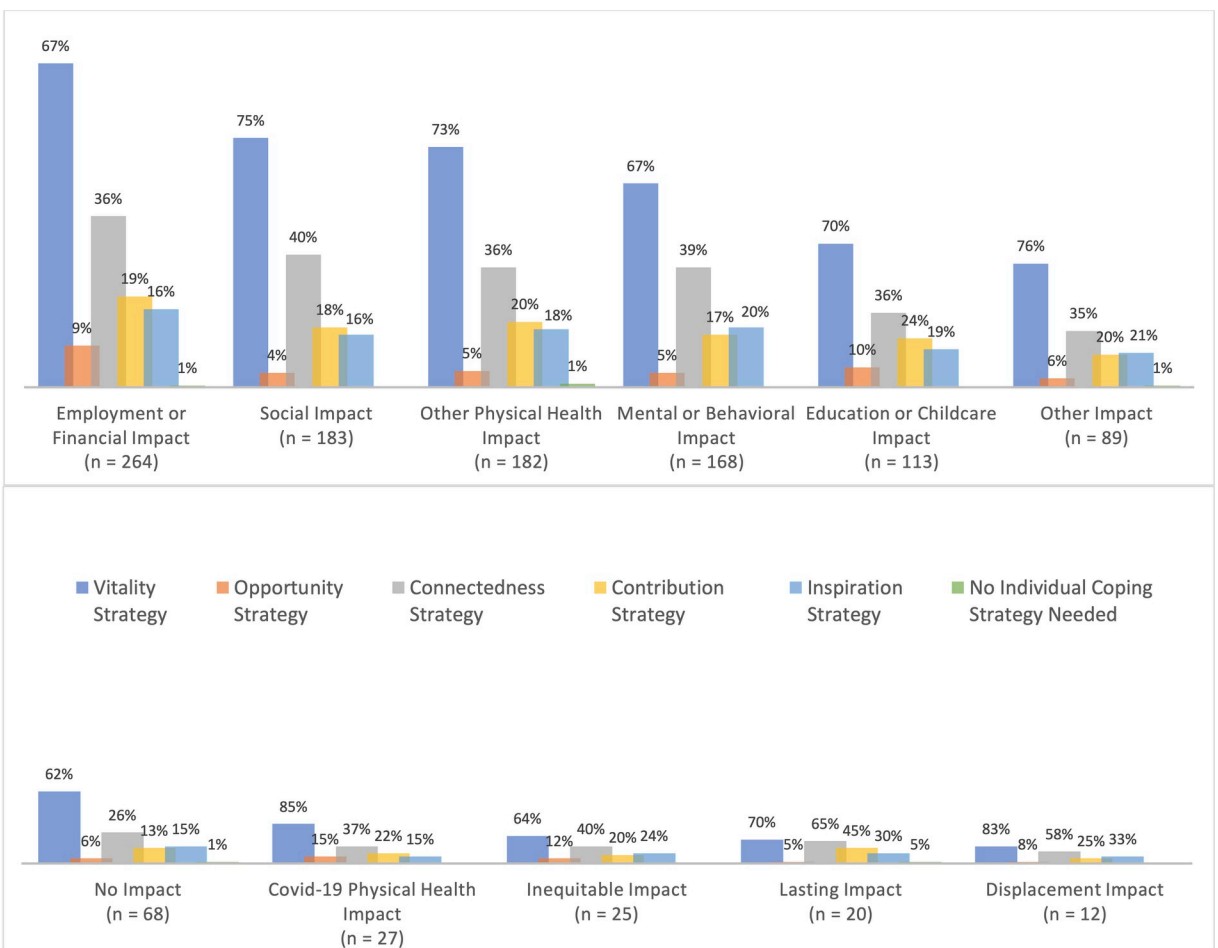

**Fig 1. Impacts of covid-19 lockdowns and associated coping strategies (N = 474).**

describe a vitality coping strategy associated with that impact. Between 62–83% of participants describing any category of impact also described engaging in a vitality coping strategy. Connectedness strategies were the second most frequently described coping strategy; again, this held true regardless of the type of impact a participant described. Between 26–65% of participants describing any category of impact also described engaging in a connectedness coping strategy. Distributions of the coping strategies engaged in less frequently also remained relatively consistent across impact categories, with some minor variations (Fig 1).

## Discussion

The King County, Washington residents who participated in this study indicated that the first several months of the COVID-19 pandemic and community-level NPIs implemented in response had impacted them, their family, and their community in a variety of ways, and described numerous coping strategies that they had employed in response to these impacts. Analysis of quantitative survey data demonstrated statistically significantly higher percentages of participants were engaged in most types of COVID-19 protective behaviors after K-12 schools, and bars, restaurants, movie theaters, and other social gathering places were ordered to close in King County on March 15th, 2020, relative to before that date. The exceptions were avoiding large gatherings, which nearly equal numbers of participants reported avoiding

before and after March 15th, as well as increased handwashing and increased hand sanitizer use, both of which fewer participants reported engaging in after March 15th. Participants responding to close-ended survey questions tended to report a greater degree of concern about the physical health or healthcare access impacts of the pandemic than the financial or social impacts; half or more of the participants indicated that they were "not concerned" about inability to work, pay bills, or retire on time.

Participants likely reported avoiding large gatherings in nearly equal numbers before and after March 15th due to guidance and/or restrictions limiting large gatherings in King County prior to that date; social distancing was recommended in King County as early as March 10th, and by March 11th, all gatherings of greater than 250 people were canceled [34]. Frequent handwashing and use of hand sanitizer were also among the first protective actions recommended by public health officials in response to the pandemic [32]. Even during the relatively early stage of the pandemic when this study was conducted, King County residents had likely heard the advice to engage in frequent handwashing and/or hand sanitizing numerous times. Thus, a lower number of participants may have reported engaging in these actions after March 15th due to message fatigue. An association between message fatigue and decreased retention of and compliance with public health guidance has been demonstrated by numerous other studies, both in the context of COVID-19 [35] and other public health emergencies [36].

Given the increasing frequency with which new infectious diseases have emerged over the past century [37], as well as the current and projected impacts of climate change on the frequency of novel zoonotic disease emergence, the likelihood of another pandemic may triple over the next 30 years [38]. Non-pharmaceutical interventions such as stay-at-home orders, shutdowns, or lockdowns reduced the morbidity and mortality due to COVID-19 [39], but appear to have resulted in unintended negative consequences for the health and wellbeing of the individuals subjected to them [40–43]. Further research is needed to fully understand the overall impact of NPIs on all domains of health and wellbeing, particularly when implemented for long periods of time. However, the documented mental health impacts of the pandemic and associated NPIs, in combination with the evidence that suggests increased psychological distress is correlated with decreased compliance with NPIs [44], indicate that identifying strategies to mitigate these unintended consequences is critical to ensure that NPIs protect the health and wellbeing of populations as intended.

In contrast to the quantitative survey findings about pandemic-related concerns, qualitative data analysis found that in open-ended narratives, participants' most frequently reported category of pandemic-related impact was an employment or financial impact, followed by social impacts. To cope with these impacts, participants most frequently described employing vitality coping strategies, which were strategies intended to support health or positive functioning, followed by connectedness strategies, which were intended to maintain or strengthen community support and interdependence.

As these results and other research on COVID-19 coping strategies demonstrate [45, 46], social connection appears to have been an important coping strategy even during the early stages of the pandemic and pandemic-related NPIs. Identifying strategies to intentionally promote social connectedness is likely a key element to mitigate the unintended negative impacts of the NPIs necessitated by future pandemics; these strategies could be proactively integrated into plans for stay-at-home orders prior to the next pandemic. However, more research is needed to understand how the concerns and coping strategies of populations subjected to NPIs evolved throughout the multiyear pandemic, and to identify how to deliver effective interventions that support and augment adaptive coping strategies. Conducting such research within a geographically bounded community can assist in identifying the physical and social assets that supported community members' adaptive coping strategies during the COVID-19

pandemic. This, in turn, will allow intervention strategies to mitigate the unintended negative impacts of future pandemic-related NPIs to build upon pre-existing community resources and social infrastructure.

Gender and age were found to be correlated with an increased likelihood of describing certain impacts and coping strategies in open-ended narratives. For instance, those who identified as female or non-binary were more likely than those who identified as male to report an impact to both physical health or mental or behavioral health, and those who identified as female were more likely than male or non-binary participants to describe employing a connectedness coping strategy. Participants below the age of 65 were much more likely to report an employment or financial impact than those 65 and older; those in the 35 to 54 years old age group were the most likely to report an education or childcare impact. However, those 65 and older were the most likely to describe employing any coping strategy where statistically significant differences were found between age groups, which included vitality, connectedness, and inspiration coping strategies. Those in an older age group also tended to report lower levels of well-being, as measured by the WHO-5 Well-being Index; those in the 65 and older age group had the lowest median WHO-5 Well-being Index Percentage Score.

Differential impacts of the pandemic and associated NPIs by gender have been documented by prior research [40–42, 47, 48]; specifically, women appear to have experienced greater impacts to mental health and well-being during the early stages of COVID-19 pandemic than men [41, 42, 47]. A survey study of over 12,000 U.S. adults conducted from March to May of 2020, which also utilized the WHO-5 Well-being Index as a measure of overall mental health, found that stay-at-home orders were associated with a decrease of 0.123 standard deviations ($p$ = 0.011) in overall mental health amongst female participants, as compared to male participants who experienced a negligible, not statistically significant decline in overall mental health [47]. Large survey studies conducted in Cyprus [41] and Austria [42] in April 2020 similarly documented higher levels of anxiety and depression amongst female participants. There is also evidence that disparities in mental health impacts between men and women continued for at least several months past the initial stay-at-home orders or lockdowns. A longitudinal study of over 70,000 participants in the U.K. conducted by Fancourt et al. from the initial lockdown in March through August 2020 found that women had higher mean anxiety and depression scores than men throughout the entire study period, though the largest gender gap was found in the first few weeks of the lockdown [40]; another U.K. study with a longer follow-up time-frame (until March of 2021) also concluded that women experienced greater reductions in overall well-being than men [48]. Though our study did not find a statistically significant difference by gender in our quantitative measure of mental health (median WHO-5 Well-being Index Percentage Scores), we did find statistically significant difference by gender in the proportion of participants describing a mental or behavioral health impact of the pandemic in their open-ended narratives, with female participants far more likely than male participants to report these impacts. Our findings, combined with the prior evidence, point to a need for effective interventions to reduce the disparate mental health impacts of pandemic-related NPIs on women, particularly during the initial stages of stay-at-home orders. These interventions could perhaps build upon the coping strategies that women were most likely to describe engaging in, vitality and connectedness coping strategies.

While less research has included non-binary individuals when comparing the mental health impacts of the pandemic by gender, a study of Spanish adults conducted from April to May 2020 included 72 participants who either identified as non-binary or neither male or female; these participants had a statistically significant higher prevalence of anxiety when compared to males and similar levels of anxiety when compared to female participants [49]. Non-binary participants in our study, similar to female participants, were more likely than male

participants to describe a mental or behavioral health impact, aligning with the findings of the Spanish study. However, as our study only included 22 non-binary participants (3% of the overall sample), further research on the pandemic's impacts on non-binary and other gender non-conforming individuals is clearly warranted, particularly given the higher baseline prevalence of anxiety, depression, and other mental health conditions in this group [50].

Prior research has also documented differential mental health and well-being impacts of the pandemic by age group [40–43]; however, the findings of other research studies, in contrast to our own, largely indicate that younger adults were more likely to exhibit worse mental health symptoms during the early pandemic [40–43]. The longitudinal study in the U.K. conducted by Fancourt et al. found higher anxiety and depression symptoms at the beginning of pandemic in younger adults as compared to older adults, though younger adults experienced more rapid improvements [40]; similarly, the early pandemic studies conducted in Cyprus [41] and Austria [42] found that younger age was associated with higher levels of anxiety and depression. Longitudinal data on how the pandemic impacted the mental health of adults in the U.S. specifically is still limited, but a U.S. Centers for Disease Control and Prevention report that tracked anxiety and depression symptoms among U.S. adults from August 2020 until February 2021 found that all age groups experienced an increase in anxiety or depression symptoms during that period, though the increase was largest among 18 to 29 year olds [43].

Given the pandemic's mental health impacts across the age spectrum, perhaps more important than understanding which age group experienced the most quantifiable impacts to mental health is uncovering the mechanisms by which these mental health impacts occurred in different age groups, and how to tailor interventions to mitigate these impacts in future pandemics. Our study documented differences in both impacts and coping strategies by age group, suggesting that certain interventions may be more effective if targeted to age-group. Younger adults experiencing the impact of disruptions to school and work routines, for instance, would likely benefit from a different intervention than older adults who have left the workforce. In the U.S., older adults are more likely to reside alone than in most other countries [51], may experience distress over higher risk of a severe case of COVID-19 if infected [52], and have pre-existing high rates of social isolation and loneliness [53]. As a result, interventions to mitigate the mental health impacts of the pandemic on this group warrant special attention. The high proportion of older adults in our study who engaged in vitality and connectedness coping strategies to deal with the pandemic's early impacts point to strategies to increase social connectedness as an important potential intervention for future pandemics.

The quantitative and qualitative data appear to diverge when examining how participants rated their level of concern about the COVID-19 pandemic's impacts to physical health, financial stability, and social isolation versus how they described the pandemic's impacts in their open-ended narratives. Whereas participants on average reported greater levels of concern over the physical health and healthcare access aspects of the pandemic than the potential financial or social impacts on the quantitative survey, they were most likely to describe employment or financial impacts and social impacts of the pandemic in their open-ended narratives. The apparent divergence in these results likely reflects both the timing of this study (the early stage of the pandemic) and the relatively high income sample that participated in our survey. Whereas close-ended survey questions only asked participants to indicate their level of concern about impacts to their own employment or financial stability, the open-ended prompt allowed participants to reflect on impacts to friends, family, and the broader community. While participants in our sample may not have had high levels of concern about their own financial security, many expressed concern over the financial security or employment impacts that other members of their community were experiencing. Additionally, when given a chance to reflect on the impacts to family members in close-ended survey questions, participants

tended to report greater concern over impacts to a family member rather than themselves. Thus, during this early stage of the pandemic, many participants' concerns focused on the health and well-being of others, rather than their own personal health and well-being. More research is needed to understand how the concerns of populations subjected to NPIs evolved over the course of the pandemic, but emerging evidence suggests that as the pandemic and associated lockdowns wore on for several years, empathy and concern for the wellbeing of others appears to have decreased as levels of psychological distress increased [44]. This decrease in empathy is in turn correlated with decreased compliance with NPIs [44, 54].

This cross-sectional study does not capture longitudinal data on how the early impacts reported by participants may have evolved, intensified, or waned throughout the pandemic. Additionally, the non-probability sampling approach is inherently prone to bias and precludes generalization of the results to all residents of King County. As an invitation to participate in the study was broadly disseminated via various University of Washington and external partner websites, social media and communications channels, we were unable to assess the total number of potential participants reached by these invitations or calculate a survey response rate. Additionally, the study survey was only offered via the online REDCap platform, potentially excluding those with low digital literacy or technology access from study participation. We noted high participation of the broader University of Washington community of faculty, staff, and students in the study, despite study team efforts to recruit participants from outside the university by partnering with local community organizations to advertise information about study participation. Participants in this study had higher median levels of income and educational attainment, and were more likely to identify as White, female, and non-Hispanic/Latinx relative to King County residents as a whole. The relatively low participation of community members who identified as non-White and/or Hispanic precluded meaningful comparisons of impacts and coping strategies by race and ethnicity. Additionally, a qualitative narrative was only provided by one Spanish-speaking individual; thus, the qualitative data primarily included the perspectives of English-speaking survey respondents. As such, the results of this study may not be representative of the early impacts of the pandemic on King County residents as a whole. Due to the time-sensitive nature of this research, we were unable to conduct external pilot testing of the study survey instrument prior to its release, indicating that some items on the survey were not validated. Our use of a directed content analysis approach to analyze study participants' qualitative narratives may have introduced bias, as this approach aligned the qualitative data with the pre-existing framework developed by Peek et al. However, our use of an open-ended question to elicit participants' narratives reduces the potential that this bias was introduced into the qualitative analysis.

## Conclusion

Through a cross-sectional survey of 793 King County, Washington residents, this study documented the protective actions, concerns, perceived impacts and associated coping strategies of participants during the first several months of the COVID-19 pandemic and community-level NPIs. Analysis of quantitative survey data demonstrated higher percentages of participants were engaged in most types of COVID-19 protective behaviors after March 15th, 2020, relative to before that date, and that participants tended to report a greater degree of concern about the physical health or healthcare access impacts of the pandemic than the financial or social impacts. Qualitative data analysis of participants' open-ended narratives, however, found the most frequently reported impact of the pandemic was an employment or financial impact, followed by social impacts. Participants most frequently described employing vitality coping strategies, which were strategies intended to support health or positive functioning, to cope

with the impacts of the pandemic, followed by connectedness strategies, which were intended to maintain or strengthen community support and interdependence.

By capturing the early impacts of the pandemic, as well as strategies employed to cope with these impacts, this study provides evidence that can inform early stage policy formation and intervention strategies to mitigate the detrimental impacts of future pandemics and the NPIs implemented in response. Future research studies could expand on this study's findings by exploring the endurance and evolution of these early impacts and coping strategies throughout the multiyear pandemic, as well as by exploring how to develop effective interventions that support and augment adaptive coping strategies.

## Acknowledgments

We would like to thank community and media partners who shared the word about our study, and those that participated in the survey.

## Author Contributions

**Conceptualization:** Julio A. Lamprea Montealegre, Tania M. Busch Isaksen, Nicole A. Errett.

**Data curation:** Kathleen Moloney, Julio A. Lamprea Montealegre.

**Formal analysis:** Kathleen Moloney, Julio A. Lamprea Montealegre, Mallory Kennedy, Megan Archer, Carlos Contreras, Daaniya Iyaz, Juliette Randazza, Javier Silva, Nicole A. Errett.

**Investigation:** Julio A. Lamprea Montealegre, Tania M. Busch Isaksen, Nicole A. Errett.

**Methodology:** Julio A. Lamprea Montealegre, Nicole A. Errett.

**Project administration:** Julio A. Lamprea Montealegre, Tania M. Busch Isaksen, Nicole A. Errett.

**Resources:** Nicole A. Errett.

**Supervision:** Tania M. Busch Isaksen, Nicole A. Errett.

**Validation:** Kathleen Moloney, Julio A. Lamprea Montealegre, Tania M. Busch Isaksen, Nicole A. Errett.

**Visualization:** Kathleen Moloney, Julio A. Lamprea Montealegre.

**Writing – original draft:** Kathleen Moloney, Nicole A. Errett.

**Writing – review & editing:** Kathleen Moloney, Julio A. Lamprea Montealegre, Tania M. Busch Isaksen, Mallory Kennedy, Megan Archer, Carlos Contreras, Daaniya Iyaz, Juliette Randazza, Javier Silva, Nicole A. Errett.

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
