## [Decision Letter · Decision Letter 0]

9 Aug 2023

PONE-D-23-05940Assessing community-level impacts of and responses to stay at home orders: the King County COVID-19 Community Study

Dear Dr. Errett,

Thank you for submitting your manuscript to PLOS ONE. After careful consideration, we feel that it has merit but does not fully meet PLOS ONE’s publication criteria as it currently stands. Therefore, we invite you to submit a revised version of the manuscript that addresses the points raised during the review process.

ACADEMIC EDITOR:

What coding method was used in your analysis? Inductive or deductive? Please, clearly state that in your methods.

What were the measures you applied to eliminate bias in your study? Please, clearly state it.

when analyzing your qualitative data, was there a time you felt that you have reached "data saturation point" where no new information was obtained based on the open ended responses provided? Please state whether or not you reached data saturation point in your analysis.

What was the rationale of putting table 1 in the methods? Codes were generated and the generated codes from the data analysis should be part of the results. I think table one should be placed appropriately under results.

Line 206. “ A majority of the participants”-please remove A, start the sentence with majority

Study population and sample size should be included in your methods

You made mention of the WHO-5 wellbeing index score in your methods. It will be good to explain what it means for readers and possibly state the site from which it was extracted from. For example: The Persian version of WHO-5 available at ( https://www.psykiatri-regionh.dk/who-5/Pages/default.aspx ).

We look forward to receiving your revised manuscript.

Kind regards,

Ayi Vandi Kwaghe, D.V.M., M.V.Sc., P.G.D.E. Ph.D., MPH

Academic Editor

PLOS ONE

Journal Requirements:

Reviewers' comments:

Reviewer's Responses to Questions

**Comments to the Author**

1. Is the manuscript technically sound, and do the data support the conclusions?

Reviewer #1: Yes

Reviewer #2: Yes

Reviewer #3: Yes

2. Has the statistical analysis been performed appropriately and rigorously? 

Reviewer #1: Yes

Reviewer #2: I Don't Know

Reviewer #3: No

3. Have the authors made all data underlying the findings in their manuscript fully available?

Reviewer #1: No

Reviewer #2: No

Reviewer #3: No

4. Is the manuscript presented in an intelligible fashion and written in standard English?

Reviewer #1: Yes

Reviewer #2: Yes

Reviewer #3: Yes

5. Review Comments to the Author

Reviewer #1: The manuscript was well written and technically sound. The authors gave attention to details with statistical analysis performed appropriately and rigorously. However, there are a few seemingly ambiguous statements and a few details that were not stated/clarifications needed.

Abstract

1. In the Abstract, line 22 "after schools and community spaces were closed" may appear ambiguous to a first-time reader. Relating them to the lock-down could help keep the reader in the right context.

Introduction

2. Lines 82 to 86 appears unclear. For example, in line 84, there seems to be a break in thought after "disruptions to the social and economic underpinnings of health and wellbeing", in which "racial and ethnic minorities, individuals with disabilities, women, and those who are precariously housed or employed" categories were stated.

Methods

3. How was the study tool validated?

4. How was the tool translated into Spanish? where there any measures taken to ensure the same meaning was communicated to respondents despite the use of different languages?

5. Were there any exclusion criteria for the study participants?

6. Not much was said about the sociodemographic context of the study area to help provide insight to the readers about the county. For example the percentage of Hispanics/Latinx, proportion of Spanish speakers etcetera

7. L158. What were the COVID-19 protective behaviors? Although these were later described in table 3 in the results section, a few categories be stated earlier would make it easier for the reader to follow

8. L192 what version of Nvivo for Mac was used?

Discussion

9. Line 376 "Differential impacts of the pandemic and associated NPIs by gender have been documented by prior

research;" should be referenced

10. Line 377 "women appear to have experienced greater impacts to mental health and well-being during the early

stages of COVID-19 pandemic than men" should be referenced.

11. Lines 375 to 401 appear to be verbose. Studies stated in lines 378 to 392 can be represented as references in

line 397 as the evidence provided by other studies. However, this is subject to the discretion of the

authors

General suggestion/observation

An upload of the manuscript's questionnaire could help readers better understand the write up (this is optional and at the discretion of the authors)

Reviewer #2: • The study is highly relevant to the current global situation.

• The title is clear and concise, indicating that the study focuses on assessing the impacts of stay-at-home orders and community responses during the COVID-19 pandemic in King County.

• The title is already informative, but it could be made even more compelling by adding a concise mention of the key findings or some intriguing aspects of the study.

• Overall, the abstract is well-structured, informative, and provides a clear understanding of the study's objectives, methods, results, and implications. It efficiently conveys the relevance and significance of the research conducted.

• Providing surveys in both languages (Spanish and English) allows for a more representative sample and reduces potential language-related biases in the responses. Additionally, providing information on the process used for translation validation would add transparency and credibility to the study.

• In the Methods section, it would be beneficial to include details on how the translation and validation of the survey tools were performed, ensuring that the Spanish version is culturally and linguistically appropriate for the target population. including both English and Spanish versions of the survey tools is commendable and demonstrate the research team's commitment to conducting a comprehensive and inclusive study in the King County community. By addressing the translation process in the manuscript, the study would reinforce the rigor and reliability of the findings across both language groups.

• In the Methods section provided, the manuscript does not explicitly mention the specific type of qualitative data analysis conducted. It only states that "qualitative content analysis methods were used to analyze free-form narrative responses." However, it does not provide further details on the specific approach or method employed for the qualitative analysis.

To ensure transparency and rigor in the study, it is crucial for the manuscript to include more information about the qualitative content analysis method. Qualitative content analysis can take various forms, such as thematic analysis, narrative analysis, or grounded theory, among others.

To enhance the method section, the manuscript should provide more details about the qualitative content analysis method used, as this is a critical aspect of the research process, especially when analyzing free-form narrative responses. Specifying the qualitative content analysis method used would be valuable for readers to better understand the study's qualitative data analysis process and the credibility of the findings.

• The manuscript does not provide information about the total number of individuals who were invited to participate in the survey through various recruitment channels. Knowing the outreach efforts and response rate would be helpful in assessing the representativeness of the sample.

To enhance the description of the sample and data collection, the manuscript could include more details about the total number of individuals who were exposed to the recruitment materials and the response rate. Also, the population and demographics of the county.

• The codes and results are almost entirely descriptive, and it should it lifted up higher conceptually into more analytic codes.

A larger part of the results is replicated in the discussion section, which was not compared to comparable findings.

• The results and concepts in the discussion section should have been supported by quotes or notes from the qualitative aspects of this study.

• There is no explicit mention of the informed consent process for the study participants. It is essential for any research involving human participants to obtain informed consent, which ensures that participants are fully aware of the study's purpose, procedures, potential risks, and benefits before agreeing to participate. Informed consent, it should be added before submission for peer review and publication. Ethical considerations, including informed consent, are critical components of any research involving human subjects and should be transparently addressed in the manuscript.

Reviewer #3: It is a very relevant topics, understanding the impacts of COVID-10. The manuscript does a great job presenting the problem and the research questions. I really enjoyed reading it. However, I have a few comments:

- In page 9, line 152, it mentions that the participants were directed to a website to participate. Would that exclude potential participants with low digital literacy, thus more vulnerable to social isolation and other impacts of during the pandemic?

- In table 3, it seems that handwashing for at least 20 seconds with soap and water, and hand sanitizer use decreased after “March 15, 2020,” is that correct? If so, how can explain these results?

- In the same table, “avoided small or mid-size gatherings” increased, while “Avoided large gatherings” remained the same. It sounds a little bit unexpected, even contradictory. Can you elaborate more in that result?

- The analyses by gender and age groups are interesting. We know the pandemic, mainly at its beginning, had a huge toll on black and racialized communities. Did the study produce any analysis by race?

6. PLOS authors have the option to publish the peer review history of their article (what does this mean?). If published, this will include your full peer review and any attached files.

Reviewer #1: No

Reviewer #2: **Yes: **Mohammed Isa Bammami

Reviewer #3: No

---

## [Author Response · Author response to Decision Letter 0]

22 Sep 2023

Response to Reviewers

Ref: PONE-D-23-05940

We very much appreciate the thoughtful critique of our manuscript, “Assessing community-level impacts of and responses to stay at home orders: the King County COVID-19 Community Study,” and the suggestions provided by the Academic Editor and each of the Reviewers. Below we respond to each of the Editor’s and Reviewers’ comments raised in the August 9th, 2023 email correspondence. Editor and Reviewer comments are provided and are followed by itemized responses. Please note every response is linked to a page and line demarcation in the revised manuscript submission.

Academic Editor:

1. What coding method was used in your analysis? Inductive or deductive? Please, clearly state that in your methods.

Thank you for the opportunity to clarify which method of coding we employed. We used a deductive coding approach, leveraging a framework for describing disaster losses and coping strategies presented in a disaster research article by Peek et al., published in 2020. We have added language to the Methods section (page 12, lines 236-241) to clarify our approach. It now reads: “To analyze the open-ended data detailing respondents’ stories about COVID-19, we used a directed content analysis approach. A codebook was developed based on the study goals, and the framework for describing disaster losses and coping strategies presented in a recently published disaster research article by Peek et al. One member of the study team applied the codebook to a sample of open-ended responses to assess the robustness of the codebook relative to the data prior to broader application by the study team.”

2. What were the measures you applied to eliminate bias in your study? Please, clearly state it.

Thank you for the suggestion to add text about measures taken to reduce bias. We have added text to the Methods section to more clearly indicate that we employed co-coding as an approach to reduce bias. However, despite these efforts, our use of a convenience sample is inherently prone to bias. We discuss this in the context of our Limitations section (see page 34, lines 908-911) to promote transparency and temper interpretation of results. In addition, we have added discussion in the Limitations section (page 35, lines 932-936) about biases that may be introduced through a directed content analysis approach. As the approach seeks to align available data with an existent theory or framework selected a priori, researchers may be more likely to find evidence supporting the existing theory or framework, rather than refuting it. This can be exacerbated when such frameworks are also used to prompt responses from participants (e.g., in the context of a key informant interview, where questions are designed to elicit responses around elements of an existing theory or framework). However, our study’s use of an open-ended question without prompts aligned with the coding framework mitigates this potential bias. 

3. When analyzing your qualitative data, was there a time you felt that you have reached "data saturation point" where no new information was obtained based on the open ended responses provided? Please state whether or not you reached data saturation point in your analysis.

Thank you for the opportunity to clarify our study methods. Thematic saturation, or the concept of diminishing returns associated with collection of new data and responses, is an approach to determining sample sizes in qualitative studies using thematic analysis approaches, particularly in small studies. As our study employed a directed content, versus thematic, analysis approach, and included a large sample size, we did not use the concept of saturation to determine when to discontinue data collection. In directed content analysis, the content of the data is assessed against a deductive framework. In our study, we reviewed the content of open-ended responses against a framework selected a priori to determine alignment of the response with different elements of the framework. We then summarized the number of responses that aligned with each element of the framework. We did not attempt to synthesize or elicit meaning in the textual data itself to identify key themes, as would be done in a conventional thematic analysis. We have clarified that our study employed a directed content analysis, versus thematic analysis, approach in the Methods section (page 12, lines 236-241), and included a reference to content analyses. As described in our response to Academic Editor comment #2, we have also added additional discussion of the limitations of this approach to our Limitations section (page 35, lines 932-936). 

4. What was the rationale of putting table 1 in the methods? Codes were generated and the generated codes from the data analysis should be part of the results. I think table one should be placed appropriately under results.

Thank you for pointing out that the prior version of Table 1, which included illustrative quotes, would be more appropriately placed in the Results section. We placed Table 1 in the Methods section because we used a deductive coding approach, leveraging a pre-existing framework for describing disaster losses and coping strategies as described in our response to Academic Editor Comment #1. However, we concur that the illustrative quotes should not be included in the Methods section. We have retained an updated version of the table, now labeled Table 2, in the Methods section, which now includes only codes and code definitions. Illustrative quotes have now been included throughout the text of the Results section (see page 22, lines 668 - 684; page 26, lines 711-714). 

5. Line 206. “ A majority of the participants”-please remove A, start the sentence with majority

Thank you for the suggested revision to this sentence. We have removed the “A” from the beginning of the sentence as suggested (page 10, line 216). 

6. Study population and sample size should be included in your methods

Thank you for the suggestion to move information about the study sample size and participant characteristics to the Methods section. We have moved the Participant Characteristics section and the table providing an overview of participant characteristics (formerly Table 2, now Table 1 in the revised manuscript) to the Methods section (pages 10-11, lines 211-221). 

7. You made mention of the WHO-5 wellbeing index score in your methods. It will be good to explain what it means for readers and possibly state the site from which it was extracted from. For example: The Persian version of WHO-5 available at ( https://www.psykiatri-regionh.dk/who-5/Pages/default.aspx ).

Thank you for bringing to our attention that the meaning of WHO-5 Well-being Index percentage scores was not well explained in our Methods section. We have added additional text to this section to explain what the percentage scores indicate, as well as inserted a link to a website from which both the Spanish and English versions on the WHO-5 questionnaire can be obtained (page 8, lines 170-176). 

Journal Requirements: 

Thank you for providing these references. We have confirmed that our revised manuscript complies with PLOS ONE’s style requirements, as outlined in the documents provided.

Thank you for the opportunity to clarify our study procedures. The University of Washington Human Subjects Division reviewed our study protocol and determined that the study qualified for exempt status. Though an exempt status waives the requirement to document informed consent, potential participants were provided an overview of the potential risks and benefits of study participation on the study webpage. The link to the study survey on this website informed participants that they were consenting to participate in the research study by answering survey questions. We have updated the Methods section of the manuscript (page 9, lines 187-191) and online submission form with this information. 

Thank you for the opportunity to clarify how other researchers may access our study’s de-identified dataset. We have created and published a project for this study on the National Science Foundation-funded DesignSafe-CI’s Data Depot Repository, and have uploaded the de-identified study dataset to the repository. In alignment with the repository's Protected Data Best Practices, this published dataset includes participants’ responses to quantitative survey questions and the results of the qualitative data analysis of participants’ written narratives (e.g., the binary ‘1’ or ‘0’ variable for each code to indicate if it was present in a participant’s narrative). Participants’ written narratives are not included in this publicly available dataset, as many contain sensitive and/or potentially identifiable information. As survey responses were collected from a small geographical area, sampling heavily from the University of Washington community, complete de-identification of participants’ narratives is not possible. The de-identified data set is available for public access via DesignSafe project PRJ-2997 (DOI: https://doi.org/10.17603/ds2-atw6-7z47).

We have reviewed our reference list for completeness and accuracy. As part of the revisions requested by the Academic Editor and the Reviewers, we have added the following additional references:

1. WHO-5 Questionnaires. https://www.psykiatri-regionh.dk/who-5/who-5-questionnaires/Pages/default.aspx. Accessed 15 Sep 2023

2. United States Census Bureau: Communications Directorate - Center for New Media QuickFacts: King County, Washington. https://www.census.gov/quickfacts/fact/table/kingcountywashington/POP010220. Accessed 14 Sep 2023

3. U.S. Census Bureau 2022 American Community Survey 1-year Estimates Subject Tables: Language Spoken at Home.

4. Hsieh H-F, Shannon SE (2005) Three approaches to qualitative content analysis. Qual Health Res 15:1277–1288

5. Varty A (2020) Coronavirus timeline: How the outbreak has unfolded. In: The Seattle Times. https://www.seattletimes.com/seattle-news/health/coronavirus-timeline-how-the-outbreak-unfolded/. Accessed 15 Sep 2023

6. Guan M, Li Y, Scoles JD, Zhu Y (2023) COVID-19 Message Fatigue: How Does It Predict Preventive Behavioral Intentions and What Types of Information are People Tired of Hearing About? Health Commun 38:1631–1640

7. Baseman JG, Revere D, Painter I, Toyoji M, Thiede H, Duchin J (2013) Public health communications and alert fatigue. BMC Health Serv Res 13:295

Reviewers' comments:

Reviewer #1: 

The manuscript was well written and technically sound. The authors gave attention to details with statistical analysis performed appropriately and rigorously. However, there are a few seemingly ambiguous statements and a few details that were not stated/clarifications needed.

We thank Reviewer #1 for their review and for positive overall feedback on our manuscript. 

Abstract

1. In the Abstract, line 22 "after schools and community spaces were closed" may appear ambiguous to a first-time reader. Relating them to the lock-down could help keep the reader in the right context.

We appreciate Reviewer #1’s point that those reading the Abstract may not immediately connect the closure of schools and community spaces to the implementation of a stay-at-home order. We have added additional text to that sentence to clarify the connection (page 3, line 51).

Introduction

2. Lines 82 to 86 appears unclear. For example, in line 84, there seems to be a break in thought after "disruptions to the social and economic underpinnings of health and wellbeing", in which "racial and ethnic minorities, individuals with disabilities, women, and those who are precariously housed or employed" categories were stated.

We thank Reviewer #1 for pointing out that this sentence was unclear. We have rephrased this sentence to improve clarity. The sentence now reads, “Racial and ethnic minorities, individuals with disabilities, women, and those who are precariously housed or employed have borne a disproportionate burden across all categories of the pandemic’s impacts, including risk of COVID-19 exposure, additional health-related consequences, and disruptions to the social and economic underpinnings of health and wellbeing” (page 6, lines 111-115). 

 Methods

3. How was the study tool validated?

We appreciate Reviewer #1’s question about the validation of our study survey instrument. Given the time-sensitive nature of the study, which was focused on capturing the real-time impacts and coping strategies implemented during the early stages of the pandemic and the associated NPIs, we did not have time to fully pilot test our study survey. However, the survey was reviewed by members of the research team with a variety of education and experience levels, including undergraduate and graduate students, to ensure accessibility and readability. Additionally, the survey included previously validated measures such as the WHO-5 Well-being Index. After the survey was built in REDCap and prior to its release, multiple members of the research team piloted taking the survey via the REDcap form. We have added a brief description to the Methods section to describe our internal survey testing process (pages 8-9, lines 181-184). Additionally, we have added text to the Limitations section to clarify that we did not have time to more thoroughly test and validate the study survey instrument (page 35, lines 930-932).

4. How was the tool translated into Spanish? where there any measures taken to ensure the same meaning was communicated to respondents despite the use of different languages?

We appreciate Reviewer #1 bringing to our attention that our translation process was not detailed in the manuscript. Julio A. Lamprea Montealegre, a co-author of this manuscript, translated all survey items without a pre-existing validated Spanish translation. Dr. Lamprea Montealegre is a native Spanish-speaker who has lived in the United States for over 15 years and is fluent and practices medicine in English. We have added text to the Methods section detailing this translation process (page 8, lines 179-181). 

5. Were there any exclusion criteria for the study participants?

We also appreciate Reviewer #1 bringing to our attention that exclusion criteria was not explicitly stated in the manuscript. Only those under 18 years of age or who were not residents of King County were excluded from study participation. We have added text to the Methods section stating this exclusion criteria (page 9, lines 202-203).

6. Not much was said about the sociodemographic context of the study area to help provide insight to the readers about the county. For example the percentage of Hispanics/Latinx, proportion of Spanish speakers etcetera

We wholeheartedly agree with Reviewer #1’s comment that an overview of the social and demographic characteristics of King County, the study area, would help readers better understand the study context. We have added an additional Study Setting section under Methods that provides a basic overview of the county, including location, race, ethnicity and language data, education, and median household income (see page 9, lines 191-198). 

7. L158. What were the COVID-19 protective behaviors? Although these were later described in table 3 in the results section, a few categories be stated earlier would make it easier for the reader to follow

We thank Reviewer #1 for pointing out that providing examples of COVID-19 protective behaviors earlier in the manuscript improves clarity. We have included a few examples of such behaviors in the sentence Reviewer #1 indicated above (page 12, lines 227-228). 

8. L192 what version of Nvivo for Mac was used?

We thank Reviewer #1 for this question. Unlike Nvivo for Windows, Nvivo for Mac does not have a version number. 

Discussion

9. Line 376 "Differential impacts of the pandemic and associated NPIs by gender have been documented by prior research;" should be referenced

We agree with Reviewer #1’s comment that the above sentence should be referenced, and have added several appropriate citations at the end of the sentence mentioned (page 30, line 815). 

10. Line 377 "women appear to have experienced greater impacts to mental health and well-being during the early stages of COVID-19 pandemic than men" should be referenced.

Just as above, we agree with Reviewer #1’s comment that this sentence should be referenced, and have added several appropriate citations at the end of the sentence mentioned (page 30, line 817). 

11. Lines 375 to 401 appear to be verbose. Studies stated in lines 378 to 392 can be represented as references in line 397 as the evidence provided by other studies. However, this is subject to the discretion of the authors

We thank Reviewer #1 for this comment. While we have retained some of the discussion about the studies referenced, as we feel this overview of the prior research will be helpful for readers unfamiliar with these studies, we have made several edits to the text to reduce the length of this section (pages 30-31, lines 824-841)

General suggestion/observation

An upload of the manuscript's questionnaire could help readers better understand the write up (this is optional and at the discretion of the authors)

We wholeheartedly agree with Reviewer #1’s suggestion to publish our study survey instrument. We have created and published a project for this study on the National Science Foundation-funded DesignSafe-CI’s Data Depot Repository, and have included the survey instrument as part of the publicly available materials that can be accessed via this project (DesignSafe project PRJ-2997; DOI: https://doi.org/10.17603/ds2-atw6-7z47).

Reviewer #2: 

1. The study is highly relevant to the current global situation.

We thank Reviewer #2 for their review, and for this comment highlighting the relevance of our study’s findings. 

2. The title is clear and concise, indicating that the study focuses on assessing the impacts of stay-at-home orders and community responses during the COVID-19 pandemic in King County.

We appreciate Reviewer #2’s comment regarding our title, and its accurate representation of the focus of our study. 

3. The title is already informative, but it could be made even more compelling by adding a concise mention of the key findings or some intriguing aspects of the study.

We appreciate Reviewer #2’s suggestion. While we agree that adding additional content about results to the title could make it more compelling, our study had numerous key findings that are difficult to summarize succinctly, due to the descriptive nature of the study. We want to be mindful of balancing the length of the title with the inclusion of multiple key findings. As such, we have decided to retain the original title. 

4. Overall, the abstract is well-structured, informative, and provides a clear understanding of the study's objectives, methods, results, and implications. It efficiently conveys the relevance and significance of the research conducted.

We greatly Reviewer #2’s comment regarding our Abstract, and their overall positive review of the manuscript. 

5. Providing surveys in both languages (Spanish and English) allows for a more representative sample and reduces potential language-related biases in the responses. Additionally, providing information on the process used for translation validation would add transparency and credibility to the study.

We appreciate Reviewer #2’s comment, and agree that our translation process should be detailed in the manuscript. As we indicated above in our response to Reviewer #1, co-author Julio A. Lamprea Montealegre, a native Spanish-speaker with a high level of English language fluency, translated all survey items without a pre-existing validated Spanish translation. We have added text to the Methods section detailing this translation process (page 8, lines 179-181).

6. In the Methods section, it would be beneficial to include details on how the translation and validation of the survey tools were performed, ensuring that the Spanish version is culturally and linguistically appropriate for the target population. including both English and Spanish versions of the survey tools is commendable and demonstrate the research team's commitment to conducting a comprehensive and inclusive study in the King County community. By addressing the translation process in the manuscript, the study would reinforce the rigor and reliability of the findings across both language groups.

We appreciate Reviewer #2’s comment, and agree that our validation process for the Spanish translation should also be detailed in the manuscript. As mentioned above, a study team member who is a native Spanish-speaker with a high level of English language fluency translated the survey. The study team also included another native Spanish speaker who had long resided in the U.S., and also had a very high level of English fluency. We have added text to the Methods section detailing this validation process for the translation (pages 8-9, lines 179-184).

7. In the Methods section provided, the manuscript does not explicitly mention the specific type of qualitative data analysis conducted. It only states that "qualitative content analysis methods were used to analyze free-form narrative responses." However, it does not provide further details on the specific approach or method employed for the qualitative analysis.

To ensure transparency and rigor in the study, it is crucial for the manuscript to include more information about the qualitative content analysis method. Qualitative content analysis can take various forms, such as thematic analysis, narrative analysis, or grounded theory, among others.

To enhance the method section, the manuscript should provide more details about the qualitative content analysis method used, as this is a critical aspect of the research process, especially when analyzing free-form narrative responses. Specifying the qualitative content analysis method used would be valuable for readers to better understand the study's qualitative data analysis process and the credibility of the findings.

We agree with Reviewer #2’s suggestion to include additional description of the qualitative analysis methods we used in our Methods section. As mentioned in response to the Academic Editor above, we used a deductive coding approach, leveraging a previously published framework for describing disaster losses and coping strategies to develop the study codebook. A directed content analysis approach was used to assess the content of the qualitative narratives against this deductive framework. We have added additional text to the Methods section (page 12, lines 236-241) as Reviewer #2 suggested to clarify our approach. 

8. The manuscript does not provide information about the total number of individuals who were invited to participate in the survey through various recruitment channels. Knowing the outreach efforts and response rate would be helpful in assessing the representativeness of the sample.

To enhance the description of the sample and data collection, the manuscript could include more details about the total number of individuals who were exposed to the recruitment materials and the response rate. Also, the population and demographics of the county.

We appreciate Reviewer #2’s comment about including a response rate. As our study survey was broadly disseminated via various University of Washington and external partner websites and social media channels, we are unable to assess the total number of potential study participants that were reached by the study participation invitations distributed via those channels (see the Sample and Data Collection section for a description of these dissemination channels). We have added text describing this inability to assess the response rate to the Limitations section (page 34, lines 911-914). In addition, we have added an additional Study Setting section to our Methods, which includes information on the total population of King County, the geographic area from which our study sample was drawn (page 9, lines 192-193). 

9. The codes and results are almost entirely descriptive, and it should it lifted up higher conceptually into more analytic codes.

We thank Reviewer #2 for their comment, and for the opportunity to further clarify our study methods. As stated in our response to the Academic Editor above, our study employed a directed content, versus thematic, analysis approach. Following this approach, we reviewed the content of participants’ qualitative narratives against a framework selected a priori to determine alignment of the response with different elements of the framework. We then summarized the number of responses that aligned with each element of the framework, rather than synthesizing or eliciting meaning in the textual data itself to identify key themes, as would be done in a conventional thematic analysis. We have clarified that our study employed a directed content analysis approach in the Methods section (page 12, lines 236-241), and included a reference to content analyses. We have also added additional discussion of the limitations of this approach to our Limitations section (page 35, lines 932-936). 

A larger part of the results is replicated in the discussion section, which was not compared to comparable findings.

We thank Reviewer #2 for their comment. In the Discussion section, we compared our study’s findings to previous research findings about the differential impacts of the COVID-19 pandemic and associated NPIs by age (pages 32-33, lines 855-882) and gender (pages 30-31, lines 814-852). We also compare our results with the previous literature on mental health, such as the importance of social connection as a key coping strategy (page 29, lines 787-800). We welcome any additional suggestions from Reviewer #2 about how to relate our findings to the previous scientific literature. 

10. The results and concepts in the discussion section should have been supported by quotes or notes from the qualitative aspects of this study.

We agree with Reviewer #2’s suggestion that discussions of qualitative data analysis results should have been accompanied by additional illustrative quotes. We have added additional quotes, which were previously included in Table 2, throughout the Results section (see page 22, lines 668 - 684; page 26, lines 711-714). 

11. There is no explicit mention of the informed consent process for the study participants. It is essential for any research involving human participants to obtain informed consent, which ensures that participants are fully aware of the study's purpose, procedures, potential risks, and benefits before agreeing to participate. Informed consent, it should be added before submission for peer review and publication. Ethical considerations, including informed consent, are critical components of any research involving human subjects and should be transparently addressed in the manuscript.

We thank Reviewer #2 for this important comment and the opportunity to clarify our study procedures. As stated in response to the Academic Editor above, the University of Washington Human Subjects Division reviewed our study protocol and determined that the study qualified for exempt status. Potential participants were provided an overview of the potential risks and benefits of study participation on the study webpage and informed that they were consenting to participate in the research study by answering survey questions. We have updated the Methods section of the manuscript (page 9, lines 187-190) with this information. 

Reviewer #3: 

1. It is a very relevant topics, understanding the impacts of COVID-10. The manuscript does a great job presenting the problem and the research questions. I really enjoyed reading it. 

We appreciate Reviewer #3’s review and overall positive feedback on our manuscript. 

2. However, I have a few comments:

In page 9, line 152, it mentions that the participants were directed to a website to participate. Would that exclude potential participants with low digital literacy, thus more vulnerable to social isolation and other impacts of during the pandemic?

We appreciate and agree with Reviewer #3’s comment, and have added additional text to the Limitations section reflecting this potential exclusion (page 34, lines 914-916). Unfortunately, due to the time-sensitive nature of this rapid response research and the lack of specific funding to support this study, we did not have the resources to distribute the survey via other methods. 

3. In table 3, it seems that handwashing for at least 20 seconds with soap and water, and hand sanitizer use decreased after “March 15, 2020,” is that correct? If so, how can explain these results?

We appreciate Reviewer #3’s thoughtful question. The observation that both handwashing and the use of hand sanitizer were reported by fewer participants after March 15th is correct. We believe this is possibly explained by message fatigue. As handwashing and the use of hand sanitizer were among the first protective actions recommended by public health officials in response to the pandemic, participants had likely heard these numerous times, even at the relatively early stages of the pandemic when the study was conducted. We have added a paragraph discussing this theory, as well as the prior literature correlating message fatigue with decreased compliance with public health guidance, to the Discussion section (pages 27-28, lines 755-767). 

4. In the same table, “avoided small or mid-size gatherings” increased, while “Avoided large gatherings” remained the same. It sounds a little bit unexpected, even contradictory. Can you elaborate more in that result?

We thank Reviewer #3 for the opportunity to reflect a bit further on this result. This result is likely explained by the timeline of public health guidance for King County, which limited gatherings incrementally. Participants likely reported avoiding large gatherings in nearly equal numbers before and after March 15th due to guidance and restrictions limiting large gatherings in King County prior to that date. For instance, social distancing was recommended in King County as early as March 10th, and by March 11th, all gatherings of greater than 250 people were canceled. We have added text discussing this in the Discussion section (page 27, lines 755-758). 

5. The analyses by gender and age groups are interesting. We know the pandemic, mainly at its beginning, had a huge toll on black and racialized communities. Did the study produce any analysis by race?

We greatly appreciate Reviewer #3’s important comment, and concur that there is a great deal of evidence documenting the disproportionate impact of the COVID-19 pandemic on BIPOC communities. In our specific study, we had relatively low participation from community members who identified as a race other than White. As noted in our Limitations section, study participants were also more likely to have higher median levels of income and educational attainment, and to identify as White or non-Hispanic/Latinx, relative to King County residents as a whole. Given the relatively low representation of BIPOC communities in our study sample and the convenience sampling approach we employed to recruit study participants, we are hesitant to make comparisons between racial or ethnic groups because we fear those results would not accurately represent the pandemic’s disproportionate impact on those populations. We have added additional text to our Limitations section noting that we were unable to make these comparisons (pages 34-35, lines 922-927).

---

## [Decision Letter · Decision Letter 1]

20 Dec 2023

Assessing community-level impacts of and responses to stay at home orders: the King County COVID-19 Community Study

PONE-D-23-05940R1

Dear Dr. Moloney,

We’re pleased to inform you that your manuscript has been judged scientifically suitable for publication and will be formally accepted for publication once it meets all outstanding technical requirements.

Kind regards,

Ayi Vandi Kwaghe, D.V.M., M.V.Sc., P.G.D.E. Ph.D., MPH

Academic Editor

PLOS ONE

Additional Editor Comments (optional):

Reviewer 3 response

Please correct COVID-10 to COVID-19

Manuscript

Introduction

Line 138, Line 141- Please write KC35 in full. You cannot start a paragraph or sentence with abbreviation or acronyms.

Line 318-141 which is supposed to be the aim/objective of the study should be the last sentence in the last paragraph of the introduction.

Line 156- REDCap; Please write the meaning in full since you are beginning a sentence or you start your sentence with “ The REDCap is a “…….

Line 170-171- English and Spanish versions of the WHO-5 questionnaire are publicly available; Please clearly state the Website after the statement for easy access to readers/researchers

Line 341-PTSD; please write the meaning in full

Line 434- NPIs such as stay-at-home orders……………. Please, start the sentence with the full meaning of NPI. Do not abbreviate in the beginning of a sentence.

Line 496-we did find statistically significantly difference........Please rephrase; did you mean “Statistically significant difference”?

Line 507-these participants had a statistically significantly? Please, correct the statement.

Line 515-516-Prior research has also documented differential mental health and well-being impacts of the pandemic by age group? Please, provide the reference.

Line516-518-“however, the findings of other research studies, in contrast to our own, largely indicate that younger adults were more likely to exhibit worse mental health symptoms during the early pandemic.” Please, provide the references of the studies in contrast to your studies after this statement.

Line 567-please delete the heading “Limitations”. Limitations of the study should be the last paragraph of your discussion.

Reviewers' comments:

Reviewer's Responses to Questions

**Comments to the Author**

1. If the authors have adequately addressed your comments raised in a previous round of review and you feel that this manuscript is now acceptable for publication, you may indicate that here to bypass the “Comments to the Author” section, enter your conflict of interest statement in the “Confidential to Editor” section, and submit your "Accept" recommendation.

Reviewer #3: All comments have been addressed

Reviewer #4: (No Response)

2. Is the manuscript technically sound, and do the data support the conclusions?

Reviewer #3: Yes

Reviewer #4: Yes

3. Has the statistical analysis been performed appropriately and rigorously? 

Reviewer #3: N/A

Reviewer #4: I Don't Know

4. Have the authors made all data underlying the findings in their manuscript fully available?

Reviewer #3: No

Reviewer #4: No

5. Is the manuscript presented in an intelligible fashion and written in standard English?

Reviewer #3: Yes

Reviewer #4: Yes

6. Review Comments to the Author

Reviewer #3: (No Response)

Reviewer #4: Thank you for giving me the opportunity to review “Assessing community-level impacts of and response to stay at home orders: the King County COVID-19 Community Study” (PONE-D-23-05940R1). This is my first time reviewing this manuscript.

The authors have done a nice job addressing most of the previous reviewers’ comments. My comments are primarily focused on the qualitative methods, analysis, and results. I am not well-versed in quantitative methods so I did not review the quantitative research and analysis. Please note that the page and line numbers that I reference in the manuscript align with the “KC3S Manuscript Revision_Clean Copy” Microsoft Word file.

Much of the information about the qualitative methods is found in two sections, “Data Analysis” (pgs. 12-13, lines 228-239) and “Results” (pg. 20, lines 302-209), which makes it difficult for a reader to understand what was done. Consider briefly describing the qualitative research in the “Methods” section, including the use of directed content analysis, questions, posed, and number of responses. I appreciate that the Survey is included in materials posted to the DesignSafe Data Depot. The authors should consider providing the qualitative questions posed as an attachment to the methods section so that readers may better under the qualitative results.

In the “Participant Characteristics” section (pg. 10, lines 204-213), I would suggest adding the number of participants or percentage of survey completers who answered the open-ended questions of the survey. This number is not found in the body of the paper until the “Results” section.

In the “Data Analysis” section on pg. 12, lines 229-231, the authors describe developing the codebook “based on the study goals, and the framework for describing disaster losses and coping strategies presented in a recently published disaster research article by Peek et al”. While the study goals are described (see, for example pg. 7, lines 141-146) the framework is not described in this manuscript. I appreciate that the authors cited Peek et al, but it would help orient readers if the authors provided a few sentences describing this framework, highlighting the parts they used for the codebook.

The first submission of the manuscript included “Table 1. Qualitative codes and illustrative quotes from KC3S survey participants.” Based on the previous review, this table was renumbered as “Table 2” and moved to the end of the Data Analysis section (see pgs. 14-15, lines 240-241). Quotations that originally appeared on the table were removed but a few are included in the “Results” section. Unfortunately, many quotations that illustrated the codebook in the original submission have been lost. These quotations would help the reader better understand the codebook. I would recommend putting them in the manuscript, perhaps as supplemental material. In addition, Table 2 includes the title “Qualitative codes and illustrative quotes from KC3S survey participants.” If the authors decide not to include the quotations, they need to remove “and illustrative quotes” from this table’s title.

The quotations in the “Results” section are inconsistently presented. For example, two quotations are in italics and quotation marks (pg. 21, lines 323-324 and pg.25, lines 361-362), while all others are presented as non-italicized text within quotation marks.

This manuscript provides a good understanding of how non-pharmaceutical interventions (masking, stay-at-home orders, etc.) led to unintended impacts such as loss of income and social isolation and it illustrates how individuals responded by employing different coping strategies (for example, using virtual technologies to connect safely with family and friends) to support health and maintain positive functioning. As pointed out in the “Results” and “Discussion” sections, this information should inform early-stage policy and intervention strategy development to address future pandemics. This manuscript also highlights research gaps, such as understanding coping strategies across a multi-year pandemic.

Overall, I recommend that this manuscript be published with the minor revisions as noted in this review.

7. PLOS authors have the option to publish the peer review history of their article (what does this mean?). If published, this will include your full peer review and any attached files.

Reviewer #3: No

Reviewer #4: No

---

## [Editor Report · Acceptance letter]

31 Jan 2024

PONE-D-23-05940R1 

PLOS ONE

Dear Dr. Moloney, 

I'm pleased to inform you that your manuscript has been deemed suitable for publication in PLOS ONE. Congratulations! Your manuscript is now being handed over to our production team.

Kind regards, 

on behalf of

Dr. Ayi Vandi Kwaghe 

Academic Editor

PLOS ONE